# FGF23 and Phosphate–Cardiovascular Toxins in CKD

**DOI:** 10.3390/toxins11110647

**Published:** 2019-11-06

**Authors:** Isabel Vogt, Dieter Haffner, Maren Leifheit-Nestler

**Affiliations:** Department of Pediatric Kidney, Liver and Metabolic Diseases Hannover Medical School, 30625 Hannover, Germany; vogt.isabel@mh-hannover.de (I.V.); haffner.dieter@mh-hannover.de (D.H.)

**Keywords:** FGF23, phosphate, chronic kidney disease, cardiovascular disease, hypertension, vascular calcification, left ventricular hypertrophy

## Abstract

Elevated levels of fibroblast growth factor 23 (FGF23) and phosphate are highly associated with increased cardiovascular disease and mortality in patients suffering from chronic kidney disease (CKD). As the kidney function declines, serum phosphate levels rise and subsequently induce the secretion of the phosphaturic hormone FGF23. In early stages of CKD, FGF23 prevents the increase of serum phosphate levels and thereby attenuates phosphate-induced vascular calcification, whereas in end-stage kidney disease, FGF23 fails to maintain phosphate homeostasis. Both hyperphosphatemia and elevated FGF23 levels promote the development of hypertension, vascular calcification, and left ventricular hypertrophy by distinct mechanisms. Therefore, FGF23 and phosphate are considered promising therapeutic targets to improve the cardiovascular outcome in CKD patients. Previous therapeutic strategies are based on dietary and pharmacological reduction of serum phosphate, and consequently FGF23 levels. However, clinical trials proving the effects on the cardiovascular outcome are lacking. Recent publications provide evidence for new promising therapeutic interventions, such as magnesium supplementation and direct targeting of phosphate and FGF receptors to prevent toxicity of FGF23 and hyperphosphatemia in CKD patients.

## 1. Introduction

The major cause of death in chronic kidney disease (CKD) patients is cardiovascular disease [1]. Besides classical risk factors like smoking, dyslipidemia, and diabetes mellitus, uremic-related mineral and bone disorders (MBD) also contribute to the increased cardiovascular morbidity and mortality in CKD patients. A central role in pathologic cardiovascular remodeling is attributed to hyperphosphatemia and highly elevated fibroblast growth factor 23 (FGF23) levels [2,3,4]. Hyperphosphatemia occurs as a consequence of a decreasing glomerular filtration rate (GFR) and is known to induce vascular calcification [5]. The phosphaturic hormone FGF23 is essential for the regulation of phosphate levels in CKD patients, but excessive FGF23 levels are also associated with left ventricular hypertrophy (LVH), cardiac fibrosis, and hypertension [6,7,8]. These controversial aspects of FGF23 in CKD patients have to be taken into account when addressing FGF23 as a therapeutic target. In this review, we summarize the current knowledge of the role of FGF23 on the phosphate homeostasis in health and CKD, as well as their contribution to cardiovascular diseases. Moreover, we discuss therapeutic strategies to lower serum phosphate and FGF23 levels and how this affects the cardiovascular outcome of CKD patients.

## 2. FGF23 and its Functions in Phosphate Homeostasis

The key role of the endocrine hormone FGF23 is to maintain phosphate homeostasis. FGF23 is mainly synthesized by osteocytes in the bone. Stimulators of FGF23 secretion are primarily 1,25-dihydroxy vitamin D (1,25(OH)2D3), parathyroid hormone (PTH), and phosphate [9,10,11]. 1,25(OH)2D3 directly activates FGF23 expression by binding to the vitamin D receptor and subsequent stimulation of the FGF23 promotor region, whereas it is unknown by which mechanisms phosphate affects FGF23 expression [12,13]. Regulation of bioactive intact FGF23 levels occurs not only at the mRNA level, but also via proteolytic cleavage of the intact FGF23 into C- and N-terminal fragments by the protease Furin. Whether the cleavage fragments still have any biological activity and induce downstream signaling has to be further examined [14].

In the kidney, physiological functions of FGF23 are mediated via binding to a complex of fibroblasts growth factor receptors (FGFRs) and its specific co-receptor Klotho [15]. FGF23 lowers the renal phosphate reabsorption by activating the FGFR1–Klotho complex. Activation of the FGFR1–Klotho complex inhibits the expression and activity of the type II sodium-dependent phosphate transporters NaPi-2a and NaPi-2c, and thereby promotes the renal phosphate excretion. Furthermore, FGF23 reduces 1,25(OH)2D3 serum levels by downregulating the renal expression of CYP27B1 encoding the 1α-hydroxylase, which inhibits the conversion of the pro-hormone 25-hydroxyvitamin D3 into its active form, and by upregulating CYP24A1, which encodes the catabolic 24-hydroxylase [16]. Decreasing renal 1,25(OH)2D3 production leads to a low abundance of sodium-dependent phosphate transporter NaPi-2b in the gastrointestinal tract, thereby limiting the phosphate uptake [17]. In the parathyroid gland, binding of FGF23 to the FGFR1–Klotho complex promotes the expression of the transcription factor early growth response protein 1 (EGR1) via ERK signaling. EGR1 inhibits the gene expression and secretion of PTH, which impairs the PTH-mediated resorption of phosphate from the bone [18].

## 3. The Role of FGF23 and Phosphate Balance in CKD Progression

### 3.1. Early Stages of CKD

In CKD patients, the mineral and bone homeostasis is disturbed owing to the gradually declining kidney function. As the glomerular filtration rate (GFR) decreases, the phosphate excretion is progressively impaired. Nevertheless, in early CKD stages, serum phosphate levels are not increased, which could be explained by higher FGF23 expression in osteocytes [19]. Recently, it was shown that high dietary phosphate uptake promotes the progression of CKD. A higher dietary phosphate load increases phosphaturia, which directly correlates with a faster decrease in GFR. The accelerated decline in GFR is possibly mediated through renal tubular injury [20]. There is also evidence that dietary phosphate uptake regulates serum FGF23 levels in the healthy population and CKD patients with moderately decreased kidney function, but the exact signaling mechanism of how phosphate stimulates FGF23 secretion is still unclear [21,22].

In early CKD stages, elevated circulating levels of FGF23 are necessary to counteract phosphate retention and maintain normal serum phosphate levels by reducing the renal phosphate reabsorption [23]. In addition, increased FGF23 lower 1,25(OH)2D3 synthesis, which promotes hypocalcemia and subsequent secondary hyperparathyroidism (sHPT) during the further course of CKD progression [24,25]. Two animal studies highlighted the importance of FGF23-regulated phosphate homeostasis in CKD. The antibody-based neutralization of FGF23 in 5/6 nephrectomized rats, a well-established experimental CKD model, increases 1,25(OH)2D3 and reduces the sHPT, but also increases serum phosphate, and thereby aortic calcification. Owing to the massive calcification, neutralization of FGF23 leads to a higher risk of mortality in this study [7]. In the second study, CKD was induced in a conditional osteoblast/osteocyte-specific FGF 23 knockout mouse by feeding it an adenine-containing diet. Compared with wild type mice, the conditional FGF23 knockout in the bone shows higher serum phosphate levels and develops a more severe renal disease and cardiac hypertrophy [8]. These studies indicate that elevated FGF23 acts protectively by preventing the development of hyperphosphatemia and calcification.

### 3.2. End-Stage Kidney Disease

In end-stage kidney disease (ESKD), the phosphaturic actions of FGF23 can no longer maintain normal serum phosphate levels. Phosphate overload and FGF23-induced 1,25(OH)2D3 deficiency contribute to the downregulation of renal Klotho expression [26]. The renal Klotho deficiency reduces the affinity of FGF23 to FGFR1. This results in the development of renal FGF23 resistance, and thus an inhibition of the FGF23-mediated reduction of phosphate reabsorption [27]. Advanced loss of the kidney function together with the renal Klotho deficiency reduce the renal phosphate excretion [3,28]. Consequently, hyperphosphatemia manifests, although FGF23 levels increase up to 100–1000-fold in patients on hemodialysis [29,30,31]. The further increase of FGF23 is not only stimulated by high serum phosphate levels, but is also a result of elevated PTH levels and the frequently used 1,25(OH)2D3 therapy [32,33,34,35]. Physiologically rising FGF23 levels inhibit PTH expression and secretion, whereas in ESKD patients, the downregulation of the FGFR1–Klotho complex in the parathyroid glands causes a resistance to FGF23 [36,37,38]. Consequently, ESKD patients commonly exhibit hyperphosphatemia, FGF23 excess, 1,25(OH)2D3 deficiency, and sHPT.

## 4. The Role of FGF23 and Phosphate in CKD-Associated Cardiovascular Diseases

High levels of serum FGF23 and phosphate are independently associated with an increase in all-cause and cardiovascular mortality in CKD patients [29,30,31,39,40,41,42,43]. Both contribute to the development of cardiovascular disease by distinct mechanisms. Therefore, FGF23 and phosphate are considered as therapeutic targets to improve CKD-associated cardiovascular morbidity. As discussed earlier, targeting FGF23 alone deteriorates the cardiovascular outcome owing to the detrimental effects of hyperphosphatemia. Thus, a detailed understanding of the molecular insights of FGF23 and hyperphosphatemia in the development of cardiovascular disease is essential to develop successful therapeutic strategies.

### 4.1. Hypertension

The prevalence of hypertension increases with decreasing GFR in CKD patients [44]. This is at least partly because of high FGF23 and phosphate levels, as several clinical studies showed an association between increased serum FGF23 or phosphate and the presence of hypertension in CKD patients, as well as in non-CKD populations [45,46,47,48,49]. The ARIC (Atherosclerosis Risk in Communities) study evaluated the correlation of serum intact FGF23 levels and the development of hypertension in a large cohort of 7948 middle-aged (45–64 years) participants without previous hypertension during a median follow-up of 5.9 years. Overall, 27% of the participants developed a hypertension, whereby the prevalence increased independently of kidney function in the highest decile of serum FGF23 [45]. In the CARDIA (Coronary Artery Risk Development in Young Adults) study, the association of C-terminal FGF23 (cFGF) levels with hypertension was analyzed in a younger (18–30 years), more multiethnic population. The C-terminal FGF23 ELISA measures the total FGF23 protein including cleaved and full length FGF23 isoforms. In total, 35.2% of 1758 participants without pre-existing hypertensive or other cardiovascular diseases developed hypertension during a ten-year period from year 20 to 30. The risk of developing hypertension was 45% greater in the highest quartile of cFGF23 levels compared with the lowest quartile [46]. These clinical studies provide evidence that FGF23 might be a causal factor in the development of hypertension in CKD patients.

Therefore, FGF23 is thought to contribute to the development of hypertension by modulating the renin–angiotensin–aldosterone system (RAAS) (Figure 1). Experimental studies indicate that FGF23-mediated 1,25(OH)2D3 deficiency activates the RAAS. Disruption of 1,25(OH)2D3 signaling in vitamin D receptor null mice promotes renal renin expression and subsequent production of the vasoconstrictor angiotensin II (AngII). Ultimately, this leads to the development of hypertension [50,51]. FGF23 might activate the RAAS also by directly suppressing the angiotensin-converting enzyme 2 (ACE2). ACE2 catalyzes the conversion of AngII into the vasodilative angiotensin I (AngI); thus, suppression of ACE2 via FGF23 also results in increased AngII levels [52,53]. Another experimental study suggests that FGF23 controls renal sodium reabsorption by regulating the abundance of Na^+^Cl^−^ co-transporter (NCC) in the distal tubule (Figure 1). Through stimulation of NCC expression, FGF23 increases renal Na^+^ retention, which consequently triggers plasma volume expansion and, consequently, hypertension [54].

Besides FGF23, a large prospective study with more than 9000 hypertensive participants showed that serum phosphate levels at the initial presentation are associated with increased systolic blood pressure (BP) at five years of follow up. However, this study did not allow any conclusions on whether serum phosphate levels directly correlate with hypertension, because only baseline phosphate levels were measured in the beginning of the study [55]. In a small study of 30 diabetic CKD and 23 non-diabetic CKD patients, serum phosphate levels correlated with increased systolic and diastolic BP in diabetic CKD, but not in non-diabetic CKD patients [48]. Recently, a prospective randomized study investigated the effects of controlled inorganic phosphate (P_i_) uptake on BP in 20 healthy young adults. Controlled high dietary phosphate uptake was associated with increased systolic and diastolic BP [49]. Both studies analyzing the direct relationship between serum phosphate and hypertension were limited owing to small cohort sizes, so further clinical studies are necessary to confirm these results.

Animal studies in normotensive and spontaneously hypertensive rats revealed that a high phosphate diet triggers hypertension [56,57,58]. Feeding a high phosphate diet was shown to increase renin expression via PTH in healthy rats, which in turn stimulated AngII, and thereby induced hypertension [56]. Moreover, activation of the sympathetic nerve system (SNA) might be involved in phosphate-related BP elevation (Figure 1). Besides other molecular mechanisms, BP is controlled by the exercise pressor reflex, which originates in exercising muscles and is modulated by metabolic and mechanical stimuli. In previous human studies, an overactive exercise pressor reflex promotes SNA activity and increased BP in hypertensive patients [59,60]. In healthy rats, feeding of a high phosphate diet induced the exercise pressor reflex function together with the renal SNA activity and BP [58]. Underlying mechanisms of how phosphate alters the exercise pressor reflex are not clarified yet.

Altogether, the clinical and experimental data emphasize that high phosphate and FGF23 levels may contribute to hypertension not only in CKD patients, but also in the general population. This shows the importance of controlled dietary phosphate intake and underlines that FGF23 and phosphate represent promising therapeutic targets in treating hypertension in CKD patients.

### 4.2. Vascular Calcification

Vascular calcification (VC) describes pathological depositions of calcium–phosphate salts on the walls of blood vessels. Owing to the disturbed mineral metabolism, VC is highly prevalent and strongly associated with an increased risk of mortality in CKD patients [61,62,63]. The involvement of elevated FGF23 in the progress of VC is controversially discussed [64,65,66,67,68,69,70]. Clinical studies of 65 hemodialysis and 142 patients with CKD stages 2–5 indicate a correlation of higher FGF23 levels and increased aortic calcification [64,65]. In contrast, Scialla et al. reported that a much larger cohort of 1501 patients with a mean eGFR of 47 mL/min/1.73 m^2^ from the CRIC (Chronic Renal Insufficiency Cohort) study did not display an association between FGF23 levels and the prevalence of coronary artery calcification [66].

The CRIC study was further supported by in vitro experiments. Stimulation with FGF23 did not augment phosphate-induced calcification, neither in human vascular smooth muscles cells (VSMCs) in dependence on the P_i_ concentration, nor in aortic rings in the presence of soluble Klotho [66]. Equally, Lindberg et al. demonstrated that FGF23 treatment had no impact on phosphate-induced VC in bovine VSMCs [67]. Other experimental studies even attribute a protective role to FGF23 regarding the progress of VC [7,8,68,69,71]. As stated by Lim et al., FGF23 diminishes calcification of human VSMCs in the presence of Klotho [68]. Recently, Chen et al. further showed that overexpression of FGF23 and Klotho in rat VSMCs attenuated phosphate-induced calcification via inhibiting Wnt7b/β-catenin signaling [71]. On the contrary, Jimbo et al. reported an enhanced phosphate-induced calcification in Klotho-overexpressing VSMCs by FGF23 [70]. A protective role of FGF23 was also demonstrated by Shalhoub et al. and White et al. as depletion of FGF23 levels in rodent CKD models leads to a more severe VC [7,8]. As CKD is a state of Klotho deficiency, there is no experimental evidence that FGF23 contributes to the progress of VC; it rather acts protectively. However, further studies are required to determine the interplay of FGF23 and Klotho levels on VC to figure out whether modulation of the FGF23–Klotho axis and downstream signaling pathways might serve as therapeutic targets in CKD patients.

In contrast to the controversially discussed role of FGF23 in the development of VC, hyperphosphatemia is strongly associated with VC (Figure 1) [72,73,74,75]. Even slightly elevated serum phosphate concentrations correlated with enhanced vascular and valvular calcification in a multiethnic study in 439 patients with moderate CKD without previous clinical cardiovascular disease [73]. Moreover, a study in 286 ESKD patients undergoing chronic dialysis treatment showed reinforced aortic arch calcification, which was highly associated with the degree of hyperphosphatemia [72]. In an ex vivo model of human vessel culture, long time exposure to phosphate resulted in increased calcification and vesicle deposition in vessels from stage 5 CKD patients compared with age-matched controls [76]. These findings are further supported by rodent studies, in which feeding of a 0.9% high phosphate diet compared with a 0.5% normal diet induced VC in nephrectomized mice [74,75].

In vitro studies in VSMCs give evidence for mechanisms underlying the phosphate-induced VC. Phosphate uptake in VSMCs occurs via type III sodium-dependent P_i_ cotransporters PiT-1 and PiT-2 [77]. Both PiT transporters exert opposing functions on VC under high phosphate conditions. PiT-1 acts pathologically by stimulating osteochondrogenic differentiation of VSMCs and release of calcium–phosphate containing vesicles, whereas PiT-2 protects against VC in VSMCs [78,79,80,81]. Elevated phosphate levels stimulate the osteochondrogenic differentiation by activating the expression of osteochondrogenic genes such as runt-related transcription factor 2 (Runx2), osteopontin (OPG), alkaline phosphatase (ALP) via bone morphogenic protein-2 (BMP-2), and Wnt/β-catenin signaling. Furthermore, phosphate-induced downregulation of the smooth muscle (SM)-specific genes SMα actin and SM22α contribute to the osteochondrogenic differentiation [82,83,84,85,86,87]. Recently, it was shown that the knockout of PiT-2 augmented aortic vascular calcification in uremic mice on a 1.5% high phosphate diet compared with a 0.5% normal phosphate diet. Additionally, PiT-2 deficient VSMCs showed enhanced matrix calcification, which might be mediated via downregulation of the osteoclastogenesis inhibitory factor osteoprotegerin [80]. Further in vivo and in vitro studies suggest that hyperphosphatemia contributes to VC via downregulation of the peroxisome proliferator-activated receptor (PPAR)γ. Treatment of VSMCs with phosphate as well as hyperphosphatemia in a CKD mouse model showed a downregulation in the expression of PPARγ and its downstream target Klotho. PPARγ agonists enhanced the expression of Klotho and inhibited VC [88,89]. Thus, to the authors’ current knowledge, the direct induction of VC results from hyperphosphatemia rather than elevated FGF23 levels in CKD patients.

### 4.3. Inflammation-Mediated Vascular Calcification

Chronic inflammation is a risk factor for increased morbidity and mortality and contributes to the development of VC in CKD [90,91]. Clinical studies showed that elevated FGF23 levels are associated with higher levels of pro-inflammatory cytokines in CKD and non-CKD populations [92,93,94].

Experimental studies demonstrated that pro-inflammatory cytokines such as interleukin (IL)-1β, IL-6, and tumor necrosis factor-α (TNF-α) directly stimulate the synthesis of FGF23, and vice versa, FGF23 directly enhances the production of pro-inflammatory cytokines. This establishment of a vicious cycle augments systemic inflammation and contributes to the progression of CKD [95,96,97,98,99]. FGF23 promotes the expression of TNF-α in the kidney and via ERK signaling in macrophages [53,98]. Possibly, the activation of TNF-α production is mediated via the FGFR1–Klotho complex, which is expressed in macrophages and the kidney. Furthermore, Singh et al. showed that FGF23 stimulates the production of IL-6 and C-reactive protein (CRP) via FGFR4-mediated activation of the calcineurin–NFAT pathway in hepatocytes of 5/6 nephrectomized rats [99]. Besides FGF23, phosphate, incorporated in secondary calciprotein particles, also directly enhances the production of TNF-α and IL-1β, and thereby contributes to VC [100,101].

Elevated levels of the pro-inflammatory cytokines TNF-α, IL-6, and IL-1β directly induce VC in VSMCs by stimulating their osteochondrogenic differentiation [100,102,103,104]. TNF-α activates nuclear factor-κB (NF-κB) signaling, which induces the expression of the osteogenic transcription factor MSX-2, and subsequently the activity of the tissue non-specific alkaline phosphatase (TNAP) [91,100,105]. Activation of TNAP contributes to the formation of hydroxyapatite crystals [106]. Furthermore, TNF-α stimulates the synthesis of IL 6, which is also known to promote the osteochondrogenic differentiation of VSMCs via BMP-2-mediated upregulation of TNAP [91,107]. Besides the direct effects, inflammatory cytokines also contribute indirectly to VC in CKD by inducing the production of reactive oxygen species (ROS) and reducing fetuin-A levels, which is an important inhibitor of calcification [108,109]. As described earlier, the direct induction of VC primarily results from hyperphosphatemia, whereas the inflammation-mediated VC is promoted by elevated FGF23 and phosphate levels.

### 4.4. Left Ventricular Hypertrophy

The prevalence of LVH is about 40% in patients with early stages of CKD and rises up to 75–80% in ESKD patients [110,111,112]. Elevated serum FGF23 levels correlate with the development of LVH not only in CKD patients, but also in the elderly population with preserved kidney function [113,114,115,116]. Likewise, LVH represents the most abundant cardiovascular disease in children with CKD. In children aged 1–21 years, the prevalence of LVH is lower, with 15–67% compared with adult CKD patients, but is also associated with elevated FGF23 levels [117,118,119].

In vitro and in vivo studies showed that FGF23 directly causes cardiac remodeling independent of BP [120,121,122]. Intramyocardial or intravenous injection of FGF23 in wild type mice triggered the development of LVH. In rat cardiomyocytes, FGF23 promotes hypertrophy via FGFR4-dependent activation of the calcineurin–nuclear factor of the activated T-cells (calcineurin-NFAT) pathway, which is known to regulate hypertrophic cardiac remodeling (Figure 1) [123]. Thereby, FGF23 binds to FGFR4 independently of its co-receptor Klotho, as it is not expressed in the heart [120]. Activation of this pathway was also found in a small retrospective case-control study with childhood-onset ESKD patients. In these patients, the presence of LVH strongly correlated with increased cardiac expression of FGF23 and Klotho deficiency. Furthermore, enhanced cardiac FGF23 expression was associated with upregulation of FGFR4 and stimulation of the calcineurin–NFAT pathway in the heart [119]. However, recent publications question whether elevated FGF23 per se contributes to cardiac remodeling [124,125,126]. In these studies, patients with osteomalacia/hypophosphatemic rickets and animal models with X-linked hypophosphatemia display high serum FGF23 levels, but no cardiac remodeling. The underlying cause might be differences in the mineral metabolism and FGF23 concentrations, together with systemic alterations such as hypertension and inflammation. In contrast to CKD, X-linked hypophosphatemia is associated with low serum phosphate levels and FGF23 elevation and Klotho deficiency are less pronounced than in CKD patients [124,125,126,127]. There is strong evidence that FGF23 directly contributes to the development of LVH, especially under uremic conditions, as in CKD. Indeed, further studies are necessary to determine whether FGF23 induces hypertrophy only under certain conditions.

Similar to FGF23, higher serum phosphate levels are associated with an increased left ventricular mass (LVM) and prevalence of LVH, independent of kidney function [128,129,130,131,132]. Baseline serum phosphate levels correlated with the development of LVH in 4005 healthy young adults of the prospective CARDIA study. Serum phosphate levels were assessed at baseline, whereas LVH was analyzed by echocardiography five years later. In total, 4.5% of participants developed LVH at five years of follow up [128]. Likewise, higher dietary phosphate uptake correlated with an increased LVM in 4494 participants without pre-existing cardiovascular disease [131]. Patients with intermediate CKD stages also exhibit an association between elevated serum phosphate levels and increased LVM [130,132]. A limitation of these clinical studies is that measurements of FGF23 levels were often not performed. Hence, the observed cardiac hypertrophy might be triggered by phosphate-induced elevations of serum FGF23 as well.

Previous animal and in vitro studies could not clarify whether phosphate directly initiates the development of LVH. Feeding of a 1.2% high phosphate diet for eight weeks increased the LVM and promoted LVH in uremic rats, but not in sham-operated animals [133,134]. However, Peri-Okonny et al. did not find any changes in the left ventricular function in wild type mice fed with a 2% phosphate diet for 12 weeks [135]. In contrast, Grabner et al. and Hu et al. demonstrated that wild type mice fed with the equivalent high phosphate diet for the same time developed LVH [121,136]. The phosphate-induced cardiac hypertrophy could be prevented by a global knockout of FGFR4 [121]. This supports the hypothesis that phosphate promotes LVH by stimulating FGF23, which is known to induce cardiac hypertrophy via FGFR4. Additionally, various 1,25(OH)2D3 concentrations in a 1.6% high phosphate chow influenced phosphate-mediated hypertrophy and fibrosis in the left ventricle. Low 1,25(OH)2D3 concentrations promoted fibrosis, whereas high concentrations augmented phosphate-induced cardiac hypertrophy [137]. In vitro supplementation of P_i_ stimulated the expression of fibrotic proteins CTGF (connective tissue growth factor) and collagen I in neonatal rat cardiac fibroblast and CTGF in cardiomyocytes, whereas hypertrophy in cardiomyocytes was not affected [136]. In contrast, phosphate treatment induced hypertrophy in cardiomyoblasts, potentially via Erk1/2 signaling [138]. Clinical and experimental studies provide evidence that hyperphosphatemia contributes to the development of LVH. In the future, it has to be clarified whether phosphate can induce cardiac hypertrophy directly or only indirectly. Possible indirect mechanisms such as the phosphate-mediated elevation of FGF23 or development of hypertension may contribute to the development of LVH in CKD patients.

## 5. Therapeutic Approaches to Inhibit FGF23- and Phosphate-Mediated Cardiovascular Disease

According to the current knowledge, elevated FGF23 levels contribute to the development of LVH, whereas hyperphosphatemia primarily induces vascular calcification. Because serum FGF23 and phosphate levels are strongly linked to each other, therapeutic approaches should target both to improve the cardiovascular outcome in CKD patients. Thus, several phosphate-lowering therapeutic strategies are pursued, which subsequently should reduce FGF23 levels.

### 5.1. Restriction of Dietary Phosphate Uptake

One of the strategies to lower FGF23 and phosphate levels is to restrict dietary phosphate intake (Figure 2). Studies in healthy subjects showed that low dietary phosphate uptake reduced FGF23 levels, while a high uptake increased serum FGF23 [21,139,140]. Tsai et al. compared the effects of a very-low phosphate diet (phosphate-to-protein ratio of 8 mg/g) and low phosphate (phosphate-to-protein ratio of 10 mg/g) diet in hemodialysis patients. Both diets similarly reduced intact FGF23 levels, whereas phosphate levels were only lowered by the very low phosphate diet [141].

The uptake of phosphate depends not only on the amount of phosphate, but also on the source. The bioavailability increases from organic plant phosphate to organic animal phosphate and, finally, additives of the food industry [142]. In uremic rats, feeding of a plant-based diet significantly reduced FGF23 levels compared with a meat-based diet, but serum phosphate levels were not altered [143]. Likewise, clinical studies in CKD patients showed a FGF23 lowering effect of vegetarian diets [144,145]. Moe et al. indicated that the vegetarian diet reduces serum phosphate levels as well. This study included a small cohort of nine participants, who received a vegetarian and meat diet with equivalent nutrients under controlled conditions [144]. In contrast, Scialla et al. did not find a correlation between the percentage of plant protein intake and serum phosphate levels in a large cohort of 2938 CKD patients. Instead of a strict control of the diet, data were collected in patient interviews [145]. Thus, the results probably differ because of the divergent study setup. Clinical studies in ESKD patients showed that restricted consumption of inorganic phosphate additives attenuates serum phosphate levels. Unfortunately, serum FGF23 was not investigated in these studies [146,147]. Moreover, a low protein diet reduces serum FGF23 and phosphate levels in non-dialysis and dialysis CKD patients [148,149]. However, reduction of total protein intake was associated with an increased mortality [148]. A general restriction of protein uptake is not a suitable therapeutic strategy to reduce serum phosphate and FGF23 levels, instead, the bioavailability of phosphate in nourishment has to be considered. Less consumption of processed food with phosphate additives and meat might have beneficial clinical outcomes. Future studies need to investigate if restricted phosphate uptake improves the general and cardiovascular outcome in CKD patients.

### 5.2. Phosphate Binder

Phosphate binders sequester phosphate, and thus prevent its gastrointestinal absorption (Figure 2). The non-calcium based phosphate binders sevelamer carbonate and ferric citrate lower FGF23 levels in CKD patients, whereas lanthanum carbonate does not show a consistent effect in controlling FGF23 levels [150,151,152,153,154,155]. In contrast, calcium-containing phosphate binders did not lower or even increased serum FGF23 levels and promoted the progression of vascular calcification (Table 1) [150,151,156]. Hence, calcium-containing phosphate binders are rather inappropriate to treat CKD patients. Clinical trials analyzing the actual cardiac outcome of a phosphate binder therapy in CKD patients are still lacking.

Recent studies indicate beneficial effects of ferric citrate treatment for CKD patients [157,158,159]. Block et al. showed in a placebo-controlled clinical trial in non-dialysis CKD patients that ferric citrate significantly reduces intact FGF23 and serum phosphate levels among patients with elevated baseline phosphate (≥4.5 mg/dL). The treatment did not affect phosphate levels in patients with lower baseline serum phosphate levels [157]. Besides controlling phosphate and FGF23 levels, ferric citrate may have beneficial effects on CKD-related anemia by increasing hemoglobin, ferritin, and transferrin saturation [158]. As shown by Francis et al., ferric citrate not only reduces FGF23 and phosphate levels, but also improves the renal and cardiac function in the Col4a3 knockout mouse model of progressive CKD. Treatment with ferric citrate reduced blood urea nitrogen levels and albuminuria, two important markers of kidney function. Col4a3 knockout mice receiving ferric citrate also showed less renal interstitial fibrosis and tubular atrophy than Col4a3 knockout control mice. Furthermore, the Col4a3 knockout mice developed severe cardiac dysfunctions, which could be attenuated by ferric citrate treatment. Without treatment, the Col4a3 knockout had an average ejection fraction of 48%, which was improved to 65% in treated animals. This might be attributed to the reduced FGF23 levels, because ferric citrate treatment decreased circulating FGF23 and the cardiac expression of FGFR4, calcineurin/NFAT, and subsequent hypertrophic target genes in the heart. Overall, ferric citrate slowed the progression of CKD and improved the survival of CKD mice. Interestingly, the protective effects of ferric citrate were higher when the Col4a3 knockout mice were treated in the early stages of CKD compared with a later onset of therapy [159]. If clinical trials can confirm findings of this animal study, ferric citrate and other non-calcium-phosphate binder represent promising drugs to improve cardiac and kidney function in CKD patients.

### 5.3. Nicotinamide

Nicotinamide (niacin, vitamin B3) reduces the dietary phosphate absorption by suppressing the expression of the intestinal phosphate transporter NaPi-2b (Figure 2) [170,171]. Several clinical studies showed that 8–12 weeks of nicotinamide treatment efficiently lowers serum phosphate levels by 12% to 34% in ESKD patients on dialysis [162,163,164,165,166]. Besides the phosphate reduction, Rao et al. demonstrated a FGF23-lowering effect of nicotinamide in a randomized, placebo-controlled trial. Nicotinamide treatment for 24 weeks decreased serum FGF23 levels by 11% among patients with CKD stages 2/3b (eGFR 30–74 mL/min/1.73 m^2^) (Table 1) [172]. More recently, Ix et al. evaluated whether nicotinamide or the phosphate binder lanthanum carbonate alone or in combination are suitable to reduce serum FGF23 and phosphate levels in stage 3b/4 CKD patients during a long-term follow-up of 12 months. The randomized, placebo-controlled study included 205 non-dialysis patients with a mean eGFR of 32 mL/min/1.73 m^2^. Neither nicotinamide or lanthanum carbonate alone, nor a combination, significantly lowered serum phosphate or FGF23 over a period of 12 months [160]. Future studies have to consider new approaches for the long-term control of FGF23 and phosphate levels in non-dialysis CKD patients. In CKD patients on dialysis, nicotinamide might have beneficial clinical outcomes owing to the reduction of serum FGF23 and phosphate levels.

### 5.4. Magnesium

In vitro studies showed that magnesium attenuates phosphate-induced calcification in VSMCs (Figure 3) [173,174]. Recently, ter Braake et al. demonstrated that secondary CPP-mediated calcification in VSMCs can be reduced by magnesium supplementation. Magnesium delayed dose-dependently the conversion of the harmless primary CPPs into the toxic secondary CPP [175]. Therefore, magnesium supplements might be a promising tool to reduce hyperphosphatemia-associated cardiovascular risk in CKD patients. The cardiovascular mortality risk decreased with increasing magnesium levels in hemodialysis patients with elevated serum phosphate levels ≥6 mg/dL [176]. High serum phosphate and low serum magnesium levels further correlated with an accelerated CKD progression during a median follow-up of 44 months in CKD patients [177].

Sakaguchi et al. examined underlying mechanisms in heminephrectomized mice that received a high phosphate/normal magnesium or high phosphate/low magnesium diet. The low magnesium diet aggravated phosphate-induced tubular injuries and interstitial fibrosis in the kidney. Moreover, low magnesium upregulated the expression of the phosphate transporter NaPi-2a in the kidney, and thereby suppressed urinary phosphate excretion [178]. Contrary to the low magnesium diet, high magnesium levels may mitigate phosphate-induced kidney injuries in CKD patients. Furthermore, magnesium citrate treatment reduced serum phosphate levels and aortic calcification in rats with adenine- and phosphate-induced renal failure. The decreased VC was associated with a downregulation of the pro-chondrogenic marker Runx2 and upregulation of SMα actin [179]. Recently, Kaesler et al. investigated the effects of magnesium carbonate or nicotinamide alone or in combination, especially on VC in 5/6 nephrectomized mice. All treatments significantly lowered serum FGF23 levels, whereas serum phosphate was only slightly reduced compared with the untreated CKD mice. Magnesium carbonate and the combined treatment reduced VC in the heart, aorta, and kidneys, while nicotinamide alone even enhanced VC. Treatment with magnesium carbonate stimulated the protein expression of the intestinal NaPi-2b and type III sodium-dependent phosphate transporter Pit-1, which could not be observed in nicotinamide or combined treated CKD mice. In the kidney, nicotinamide and magnesium carbonate alone induced NaPi-2b expression; this was abolished by the combined treatment [180]. Against the therapeutic goal, increased expression of the phosphate transporters could enhance the phosphate absorption in the intestine and reabsorption in the kidney. Thus, a combined treatment probably is most suitable to reduce VC and phosphate intake in CKD patients.

A preliminary clinical trial including 125 participants showed that oral magnesium oxide slowed the progression of coronary artery calcification in stage 3/4 CKD patients during a follow-up of two years (Table 1). Thereby, the dropout rate was 10% higher in the magnesium oxide treated group compared with the control group, mainly because of diarrhea. Treatment with magnesium oxide did not influence serum phosphate levels and FGF23 levels were not measured in this study [167]. Additionally, Bressendorff et al. showed that higher dialysate magnesium increased the conversion time of primary CPP into secondary CPP in ESKD patients undergoing dialysis. The conversion time was measured by the T_50_ test, which indicates the risk of VC. Thereby, a prolonged conversion time indicates a lower risk of VC, because VC is only mediated by the secondary CPP. High dialysate magnesium also reduced serum phosphate levels [168]. The experimental and clinical studies provide evidence that magnesium supplementation is an appropriate strategy to reduce the progression of VC in CKD patients and might help to control FGF23 and phosphate levels. Hence, future studies should investigate if magnesium supplementation can reduce the cardiovascular mortality in CKD patients.

### 5.5. Other Approaches

Another approach to control serum phosphate levels is to target the phosphate absorption and reabsorption directly using phosphate transporter inhibitors (Figure 2). NaPi-2b-deficient uremic mice exhibit lower serum phosphate and FGF23 levels than wild type uremic mice [181]. However, Larsson et al. showed that two weeks’ treatment of ESKD patients with the NaPi-2b inhibitor ASP3325 failed to reduce serum phosphate levels (Table 1) [169]. Recently, Thomas et al. assessed the effects of a NaPi-2a inhibitor in healthy and uremic mice. In both mouse models, the NaPi-2a inhibitor increased the excretion of phosphate in a dose-dependent manner and reduced plasma phosphate and PTH levels [182]. Clinical studies will have to investigate the outcome of NaPi-2a inhibition in CKD patients. According to the previous studies, promoting phosphate excretion might be more efficient than reducing the absorption to control the serum phosphate level. It is also conceivable that a combined treatment, which inhibits both intestinal absorption and renal reabsorption, could have the most beneficial effect for CKD patients.

Experimental studies showed that neutralization or knockout of FGF23 worsens the cardiovascular outcome in uremic mice models owing to phosphate-induced calcification [7,8]. A general FGF23 inhibition without targeting elevated phosphate levels is probably detrimental for CKD patients. Accordingly, therapeutic strategies should specifically prevent the harmful off-target effects of FGF23 without compromising its phosphaturic actions (Figure 3). Administration of a global FGFR blocker attenuated the development of LVH in a CKD mouse model [120,122]. However, this might induce adverse side effects in the kidney. A more specific FGFR4 antibody inhibited FGF23-induced hypertrophy of cardiomyocytes in vitro. Ablation of FGFR4 in mice prevented the development of LVH at high serum FGF23 levels [121]. The direct targeting of FGFR4 is a promising approach to inhibit FGF23-mediated off-target effects in CKD patients.

Recently, Dussold et al. demonstrated the osteocyte protein dentin matrix protein 1 (DMP1) as a feasible therapeutic target to control FGF23 levels and improve cardiac and bone health in CKD patients. The Col4a4 knockout CKD mouse model displayed reduced levels of DMP1 compared with wild type mice. Genetic and pharmacological administration of DMP1 lowered serum FGF23 levels, prevented development of LVH, and corrected bone mass in Col4a4 knockout mice. Despite the positive effects, DMP1 also enhanced hyperphosphatemia [183]. Nevertheless, together with a phosphate lowering therapy, DMP1 might represent a promising target to improve the cardiac function and bone health in CKD patients.

## 6. Conclusions

Elevated FGF23 levels and hyperphosphatemia are risk factors for the excessively increased cardiovascular mortality in CKD patients. FGF23 can act directly on the heart and promote LVH. Apart from that, FGF23 plays a key role in controlling serum phosphate levels to attenuate phosphate-induced VC and hypertension. Treatment of FGF23 levels alone is inappropriate for CKD patients because it reinforces hyperphosphatemia and the associated toxic effects. Hence, former therapeutic strategies focus mainly on phosphate-lowering approaches that in consequence will also reduce FGF23 levels. Many studies demonstrated that dietary phosphate restriction, phosphate binders, or nicotinamide reduce phosphate and FGF23 levels, but studies showing the actual outcome on the cardiovascular system are missing. Recent experimental studies provide evidence for further promising therapeutic strategies, such as direct targeting of phosphate or FGFRs or magnesium supplementation to prevent toxicity of FGF23 and hyperphosphatemia in CKD patients.

## Figures and Tables

**Figure 1 toxins-11-00647-f001:**
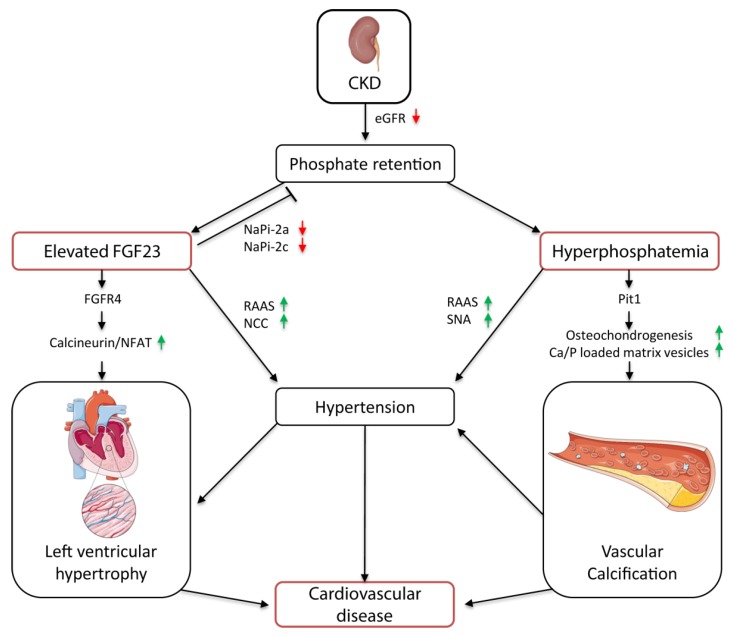
Cardiovascular pathomechanisms of elevated FGF23 and hyperphosphatemia in chronic kidney disease (CKD). The phosphate excretion attenuates with declining kidney function. Resulting FGF23 elevation counteracts the phosphate retention by downregulating NaPi-2a/c in the kidney, but also directly promotes LVH via FGFR4–calcineruin–NFAT signaling. FGF23 induces hypertension by activation of the RAAS and NCC expression. Hyperphosphatemia stimulates the osteochondrogenic differentiation and release of Ca/P loaded vesicles in VSMCs via Pit-1, and thereby induces VC. Phosphate-induced vascular calcification (VC) together with activation of the RAAS and SNA contribute to the development of hypertension. eGFR, estimated glomerulus filtration rate; NaPi-2a/c, type IIa/c sodium-dependent phosphate transporter; FGFR4, fibroblast growth factor receptor 4; NFAT, nuclear factor of activated T-cells; RAAS, renin-angiotensin-aldosterone system; NCC, Na^+^Cl^−^ co-transporter; SNA, sympathetic nerve system; VC, vascular calcification.

**Figure 2 toxins-11-00647-f002:**
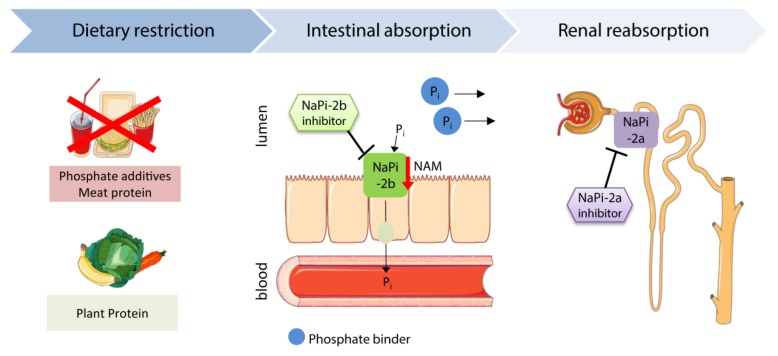
Clinical and experimental therapeutic strategies to target phosphate homeostasis in CKD. Dietary phosphate restriction and inhibition of the intestinal phosphate absorption by phosphate binders and NAM are common phosphate-lowering treatments in CKD patients. Experimental studies investigate the direct targeting of the intestinal phosphate absorption and renal reabsorption by NaPi-2b or NaPi-2a inhibitors. NaPi-2a/b, type IIa/b sodium-dependent phosphate transporter; NAM, nicotinamide.

**Figure 3 toxins-11-00647-f003:**
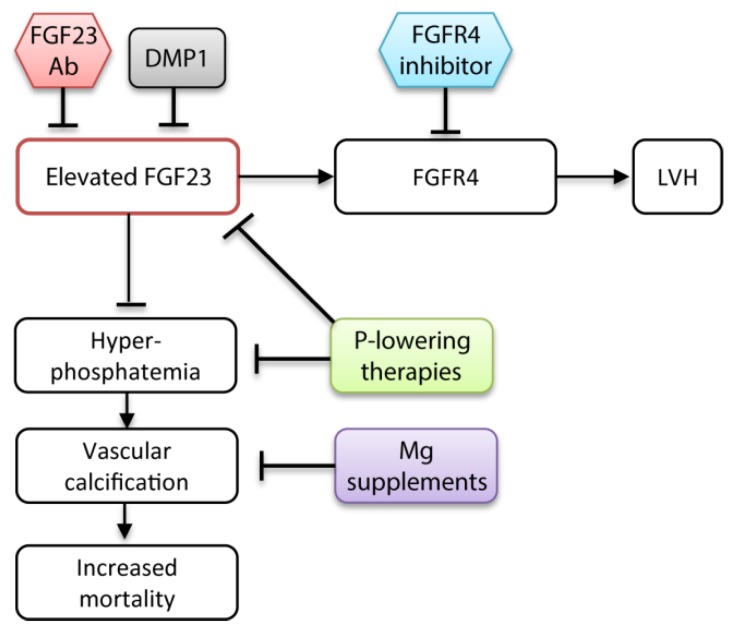
Potential treatments to target FGF23 levels and cardiovascular disease in CKD. Administration of the FGF23 Ab or DMP1 reduces FGF23 levels, but also leads to more severe VC and increased mortality. Thus, application of FGF23-reducing therapies is only conceivable together with a phosphate-lowering therapy or magnesium supplements that decrease phosphate-induced VC. The FGFR4 inhibitor specifically blocks the FGF23-mediated development of LVH. FGF23 Ab, fibroblast growth factor 23 antibody; DMP1, dentin matrix protein 1; FGFR4, fibroblast growth factor receptor 4; Mg, magnesium; LVH, left ventricular hypertrophy.

**Table 1 toxins-11-00647-t001:** Advantages and disadvantages of clinically tested phosphate- and fibroblast growth factor receptor 23 (FGF23)-lowering therapies.

Class of drug	Drug	Advantages	Disadvantages
Non-calcium phosphate binders	Sevelamer	Lower intact FGF23 and urinary Pi secretion, hypolipidemic [150,151,152]	GI side effects, high pill burden, unchanged serum Pi [150,151,152]
	Lanthanum carbonate	Good GI tolerance, lower C-term FGF23 [154]	Unchanged serum Pi and intact FGF23 [151,155,160], low solubility: tissue accumulation might cause long-term toxicity
	Ferric citrate	Lower serum Pi and intact FGF23, increase hemoglobin and ferritin [153,157,158]	Mild GI side effects [153,157,158]
Calcium-based phosphate binders	Calcium carbonate/acetate	Controlled [150,151] or lower [161] serum Pi	Hypercalcemia, VC, unchanged or increased intact FGF23 [150,151]
Nicotinamide		Lower serum Pi and intact FGF23 in dialysis patients [162,163,164]	Unchanged serum Pi and intact FGF23 in non-dialysis patients [160], mild GI side effects, at high doses hepatotoxic/thrombocytopenia [162,163,164,165,166]
Magnesium supplements	Oral magnesium oxide	Slower progression of VC [167]	Unchanged serum Pi, FGF23 not measured, Mild GI side effects [167]
	Higher dialysate magnesium	Increased conversion time from primary CPP to secondary CPP (T50 test, lower serum Pi [168]	Unchanged serum Pi, FGF23 not measured [168]
Pi-transporter inhibitor	NaPi-2b inhibitor		Not effective in reducing serum Pi [169]

GI, gastrointestinal; VC, vascular calcification; CPP, calciprotein particles; NaPi-2a, type IIa/b sodium-dependent phosphate transporter.

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
