# Peer review of "FGF23 and Phosphate–Cardiovascular Toxins in CKD"

_toxins, 2019, doi:10.3390/toxins11110647_

Round 1
Reviewer 1 Report
The authors should be commended for investigating the important role of FGF23 in the phosphate homeostasis in health and CKD, as well as their contribution to cardiovascular diseases.
This is an area that requires further exploration. The manuscript will benefit from addressing a few major issues outlined below.
The title “FGF23 – Cardiovascular Toxin and Protector of Hyperphosphatemia in CKD” indicates protective role of FGF23 but the article is more about development of hypertension, vascular calcification and left ventricular hypertrophy caused by FGF23 associated mechanisms.
Several FGF23-lowering therapeutic strategies are proposed; “5. Therapeutic approaches”. This also seems inadequate to proposed in title protective role of FGF23. In previous studies an elevated level of FGF-23 was an independent risk factor for mortality in population of patients with chronic kidney disease stages 2 through 4. The results of the previous studies confirmed and experimental studies support the hypothesis of direct toxicity of FGF-23. The article does not clearly lay out a rationale for how this study fills a critical gap in the literature
The paper needs revision in the naming of the section. There is not a clear conceptual model. Section 4. “CKD-associated cardiovascular diseases” - Could be picking up, e.g. “Fibroblast growth factor 23 (FGF23) and its functions in CKD-associated cardiovascular diseases” ?
Author Response
Comment 1: The title “FGF23 – Cardiovascular Toxin and Protector of Hyperphosphatemia in CKD” indicates protective role of FGF23 but the article is more about development of hypertension, vascular calcification and left ventricular hypertrophy caused by FGF23 associated mechanisms.
Response 1: We agree with the Reviewer and changed the title to “FGF23 and Phosphate - Cardiovascular Toxins in CKD” and hope that the new title fits better to the focus of the review regarding FGF23- and phosphate-mediated mechanisms in the development of CKD-associated cardiovascular diseases.
Comment 2: Several FGF23-lowering therapeutic strategies are proposed; “5. Therapeutic approaches”. This also seems inadequate to proposed in title protective role of FGF23. In previous studies an elevated level of FGF23 was an independent risk factor for mortality in population of patients with chronic kidney disease stages 2 through 4. The results of the previous studies confirmed and experimental studies support the hypothesis of direct toxicity of FGF23. The article does not clearly lay out a rationale for how this study fills a critical gap in the literature.
Response 2: To the best of our knowledge, many reviews focus on FGF23 or phosphate, whereas this article focuses directly on the interplay of FGF23 and phosphate in the context of the development of cardiovascular disease in CKD, and the importance of that for the decision of suitable therapeutically strategies. This is an important research field because CKD-associated CVD, with FGF23 and phosphate probably among the main drivers, still cannot be sufficiently be treated. The present review is an update on FGF23- and phosphate-mediated mechanisms in the development of CVD and new therapeutic strategies. Here, we included recently published clinical studies, mainly from 2018 and 2019, regarding the correlation of FGF23/phosphate and hypertension (please see references 46, 47, and 49). An addition, publications about new phosphate-mediated mechanisms in vascular calcification including involvement of the Wnt/b-catenin pathway (reference 71), downregulation of PPARγ (references 88 and 89), and the protective effects of PiT-2 (reference 80).
As suggested by Reviewer 2, we added a new paragraph about the contribution of FGF23 and phosphate to inflammation-mediated calcification, which is a highly discussed topic and demonstrated by rising amount of publications (references 94, 96, 97, 105, 107, and 108).
Another important issue is the current discussion whether FGF23 per se induces LVH or only under certain circumstances as in CKD. This is highly relevant for the understanding of underlying mechanisms to improve therapeutic interventions (references 125-128).
Furthermore, we intensively discussed new therapeutically strategies, in brief, listed below:
- many new clinical and experimental studies of ferric citrate (references 158-160) including one study, which shows for the first time, that a phosphate binder improves renal and cardiac function in an experimental study (reference 160)
- long-term study combined treatment with lanthanum carbonate and nicotinamide in non-dialysis patients (reference 161)
- magnesium supplements to reduce vascular calcification (references 168, 169, 176, 179-181)
- NaPi inhibitors (reference 170 and 183)
Overall, we think the above-mentioned topics summarize the current literature as much as possible.
Comment 3:The paper needs revision in the naming of the section. There is not a clear conceptual model. Section 4. “CKD-associated cardiovascular diseases” – Could be picking up, e.g. “Fibroblast growth factor 23 (FGF23) and its functions in CKD-associated cardiovascular diseases”?
Response 3: We thank the Reviewer for this suggestion and edited the naming of almost all sections in the revised manuscript, e.g.
2. FGF23 and its functions in phosphate homeostasis
3. The role of FGF23 and phosphate balance in CKD progression
4. The role of FGF23 and phosphate in CKD-associated cardiovascular diseases
5. Therapeutic approaches to inhibit FGF23- and phosphate-mediated cardiovascular disease
Reviewer 2 Report
This is a review paper dealing with the role of FGF23 in CKD as both a potential cardiovascular toxin and, at the same time, a protector from hyperphosphatemia.
This is a very current topic and the authors made a big effort to cover the most relevant part of the (huge) number of information present in the literature.
However , though I am well aware of the basic limitation in the attempt of being comprehensive of all the relevant available information in such a big field of research, I would suggest even to add some missing information and to make some minor corrections.
Given the continuous increase in the publications dealing with the connection between FGF23 and inflammation, which is one of the main drivers for vascular calcification occurrence in CKD patients, the Authors should add a dedicated paragraph to this highly relevant topic Section 2, “Role of FGF23... “; in the first paragraph, the Authors state that FGF23 is cleaved into inactive C- and N-terminal fragments. In fact, we are as yet not completely sure that these fragments are completely inactive. Section 5.2; the Authors report that Lanthanum carbonate does not show a consistent effect in controlling FGF23. However most of the papers quoted by the Authors demonstrate that FGF23 (at least C-terminal FGF23) is reduced by this phosphate binder (PB). Section 5.2; the authors do not report on some potential undesired effects associated to a prolonged use of some phosphate binders (e.g. unwanted increase in the ferritin levels with iron-based PBs; hemodynamic or gastrointestinal effects associate with NAM. Section 5.2; the Authors should comment on the global effect of magnesium on phosphorus balance, given the opposite effects at intestinal and tubular Pi transport systems. Much of the reported effects produced by the different interventions for controlling FGF23 and phosphorus levels in CKD patients could be summarized in a table , where it could be also reported the off-target effects of the different drugs, if present. By this way, space could be saved in the text of the manuscript.Author Response
Comment 1: Given the continuous increase in the publications dealing with the connection between FGF23 and inflammation, which is one of the main drivers for vascular calcification occurrence in CKD patients, the Authors should add a dedicated paragraph to this highly relevant topic.
Response 1: We thank the Reviewer for this excellent suggestion. The contribution of FGF23 and phosphate to inflammation-mediated calcification is a highly discussed topic with rising amount of publications. Therefore, we include a new paragraph 4.3 “Inflammation-mediated vascular calcification” in the revised manuscript including references 94, 96, 97, 105, 107 and 108.
Comment 2:Section 2 “Role of FGF23…”, in the first paragraph, the Authors state that FGF23 is cleaved into inactive C- and N-terminal fragments. In fact, we are as yet not completely sure that these fragments are completely inactive.
Response 2:The Reviewer is right. We therefore edited the paragraph accordingly in the revised manuscript: “If the cleavage fragments still have any biological activity and induce downstream signaling has to be further examined (reference 14).”
Comment 3:Section 5.2; the Authors report that Lanthanum carbonate does not show a consistent effect in controlling FGF23. However most of the papers quoted by the Authors demonstrate that FGF23 (at least C-terminal FGF23) is reduced by this phosphate binder (PB). The Authors do not report on some potential undesired effects associated to a prolonged use of some phosphate binders (e.g. unwanted increase in ferritin levels with iron-based PBs, hemodynamic or gastrointestinal effects associated with NAM. The Authors should comment on the global effect of magnesium on phosphorous balance given the opposite effects at intestinal and tubular Pi transport systems. Much of the reported effects produced by the different interventions for controlling FGF23 and phosphorous levels in CKD patients could be summarized in a table, where it could be also reported the off-target effects of the different drugs, if present.
Response 3:We took the suggestion of the Reviewer and include a Table 1in Section 5, summarizing the clinically tested FGF23- and phosphate-lowering therapies and their advantages and disadvantages in order to give a better overview of the different treatments and their side effects.
Additionally, we edited different comments suggested by the Reviewer in Section 5.4 “Magnesium”.
Reviewer 3 Report
This is an interesting review about the effect of elevated levels of FGF23 and phosphate in CKD patients on cardiovascular system.
Review is update and full, it is well written with clear concepts.
In my opinion there are only some minor considerations that could be led to improve it.
- Title maybe is not very precise due to during end stage of CKD FGF23 does not protect from hyperphosphatemia and there is not phosphaturia. One option could be “FGF23 – Cardiovascular Toxin vs phosphaturic hormone” or similar.
- Page 2 line 91. It would be interesting to mention some causes whereby renal Klotho is decreased.
- Page 3 line 96. After references 34-36 it would be interesting to mention the renal effect of FGF23 on FGFR1 and how FGF23 resistance is also produced at renal level through Klotho downregulation induced by phosphate overload.
- In section 4.2 related to Vascular Calcification and P it would be necessary to mention the relation between high levels of phosphate and the activation of pro-osteogenic pathways such as BMP2, Wnt/b-catenin or Notch signalling.
Similarly, the review will improve including a new section relate to FGF23/high phosphate and inflammation and CVD progression.
- Page 9 line 387. In the section relate to Magnesium should be included works describing the beneficial effects of Magnesium supplementation on CVD associated to CKD. In addition, there are some studies where it is demonstrated that dietary magnesium supplementation protects and even revers VC in animal models of uremia.
Minor comments:
- Page 2 line 72 delete space among “populationandand ”.
- Page 5 , line 195 name Shaloubad is wrong and it must be change by Shalboub.
- Page 7 line 277 Abbreviation CTGF should be described (Connective tissue growth factor).
Author Response
Comment 1: Title maybe is not very precise due to during end stage of CKD FGF23 does not protect from hyperphosphatemia and there is not phosphaturia. One option could be “FGF23 – Cardiovascular Toxin vs phosphaturic hormone” or similar.
Response 1: We agree with the Reviewer and as also suggested by Reviewer 1, we changed the title to “FGF23 and Phosphate - Cardiovascular Toxins in CKD” and hope that the new title fits better to the focus of the review.
Comment 2:Page 2, line 91. It would be interesting to mention some causes whereby renal Klotho is decreased.
Response 2: In order to point out renal Klotho decreasing factors, we added the following sentence in the revised manuscript, page 2/3, lines 92-93 “Phosphate overload and FGF23-induced 1,25(OH)2D3 deficiency contribute to the downregulation of renal Klotho expression (reference 26)”.
Comment 3:Page 3, line 96. After references 34-36 it would be interesting to mention the renal effects of FGF23 on FGFR1 and how FGF23 resistance is also produced at renal levels through Klotho downregulation induced by phosphate overload.
Response 3: We added the following sentence to depict the interplay of renal Klotho deficiency and the development of FGF23 resistance, page 3 lines 93-95“The renal Klotho deficiency reduces the affinity of FGF23 to FGFR1. This results in the development of renal FGF23 resistance and thus an inhibition of the FGF23-mediated reduction of phosphate reabsorption (reference 27)”
Comment 4:In Section 4.2 related to Vascular Calcification an P it would be necessary to mention the relation between high levels of phosphate and the activation of pro-osteogenic pathways such as BMP2, Wnt/b-catenin or Notch signaling.
Response 4: This is a good point.We now discuss the relation between P and osteochondrogenic differentiations including pro-osteogenic pathways in the revised manuscript Section 4.2, page 6, lines 224-229.
Comment 5:Similarly, the review will improve including a new section related to FGF23/high phosphate and inflammation and CVD progression.
Response 5: We fully agree with the Reviewer and as also suggested by Reviewer 2, we included a new section 4.3 “Inflammation-mediated vascular calcification” in the revised manuscript including references 94, 96, 97, 105, 107, and 108.
Comment 6:Page 9, line 387. In the section related to Magnesium should be included works describing the beneficial effects of Magnesium supplementation on CVD associated to CKD. In addition, there are some studies where it is demonstrated that dietary magnesium supplementation protects and even reverses VC in animal models of uremia.
Response 6: The Reviewers suggestions are very helpful to make the review more complete. We now include additional studies about the beneficial effects of Magnesium supplementation in the revised manuscript page 11.
Comment 7:Page 2, line 72 delete space among “populationandand”. Page 5, line 195 name Shaloubad is wrong and it must be change by Shalhoub. Page 7, line 277 Abbreviation CTGF should be described (Connective tissue growth factor).
Response 7: We thank the Reviewer for these hints and edited the manuscript accordingly.
Round 2
Reviewer 2 Report
no further comment